# Reactive Attachment Disorder and Its Relationship to Psychopathology: A Systematic Review

**DOI:** 10.3390/children10121892

**Published:** 2023-12-06

**Authors:** Florencia Talmón-Knuser, Francisco González-Sala, Laura Lacomba-Trejo, Paula Samper-García

**Affiliations:** 1Department of Developmental Psychology and Education, Faculty of Health Science, Catholic University of Uruguay, Montevideo 11600, Uruguay; florencia.talmonk@ucu.edu.uy; 2Department of Developmental and Educational Psychology, Faculty of Psychology and Speech Therapy, Universitat de València, 46010 Valencia, Spain; francisco.gonzalez-sala@uv.es (F.G.-S.); laura.lacomba@uv.es (L.L.-T.); 3Department of Basic Psychology, Faculty of Psychology and Speech Therapy, Universitat de València, 46010 Valencia, Spain

**Keywords:** reactive attachment disorder, children, adolescents, internalising problems, externalising problems, systematic review

## Abstract

This study aimed to determine, through a systematic review, the relationship between Reactive Attachment Disorder (RAD) and the presence of psychopathology in children and adolescents, and to determine the existence of differences in terms of internalising and externalising psychological problems between the RAD group and groups with other disorders or with typical development. Following the PRISMA methodology, a search was carried out in the Web of Science, PubMed and Scopus databases. The search yielded 770 results, of which only 25 met the inclusion criteria. The results indicate a relationship between the presence of RAD and/or disinhibited social engagement disorder (DSED), with the presence of internalising and externalising problems. These difficulties are more present in children with RAD compared to children without personal difficulties, or children with DSED, children with autism, children with intellectual disabilities or children with hyperactivity. It can be concluded that the presence of RAD has negative consequences on the mental health of children and adolescents, with these being greater in the inhibited group than in the disinhibited group, and with respect to children with autism or hyperactivity.

## 1. Introduction

The quality of the interactions established in the first years of human life is crucial and plays a significant role in the way human beings bond affectively and adapt personally and socially during adolescence and adulthood [1]. 

Attachment is the emotional connection between a baby and its caregiver, also called the attachment figure [2]. According to [3], the first attachments are usually established at seven months of age and are often conditioned by the primary caregiver’s responses to the child’s needs, establishing an internal model of representation of the child’s self, the caregiver and the relationship between the two [4]. 

Reactive Attachment Disorder (RAD hereafter) is characterised by difficulties in forming emotional attachments to others, a reduced ability to experience well-being and a fluctuation in mood that is apparently unexplained [5]. These alterations should appear after the age of 9 months and before the age of 5 years, having as their origin the experience of neglectful and unstable care [6,7,8,9]. 

In earlier versions of the DSM [10], RAD is divided into two types: inhibited (RAD-I hereafter), characterised by emotionally withdrawn behaviours, fear of other people and selectivity in the choice of attachment figure; and disinhibited (DSED hereafter), characterised by excessive sociability and an indiscriminate response in showing attachment to other people, even those unknown to the child [11,12].

However, the subtypes are now understood to be distinct diagnostic entities, with RAD corresponding to the inhibited form in DSM-IV and DSED to the disinhibited form [6,13]. For [13], RAD is related, to a greater extent, to poorer care in the first five years of life compared to DSED, where this relationship is not as evident.

The presence of RAD or DSED has been associated with being male; with other comorbid disorders; with parental mental health problems [14]; with having been exposed to a greater number of traumatic events or adverse childhood experiences [8]; with having had an early separation experience as a consequence of protective measures being taken [15], mainly with regards to DSED symptomatology when the change of caregivers occurs for the first time between 7 and 24 months, regardless of the severity of maltreatment, the age of adoption and the number of family changes [16]; with having experienced poor caregiving in infancy, primarily in the development of RAD-I symptomatology; and with insecure or disorganised attachments in the case of inhibited and disinhibited symptomatology [13].

Focusing on RAD, according to the DSM-5-TR [6], comorbidity with some developmental difficulties is common, especially cognitive and language difficulties, stereotypies, internalising and externalising problems such as attention deficit hyperactivity disorder, conditions that have been associated with neglect and abuse experiences [17,18] and institutional care or frequent changes of caregivers [3,18,19,20,21].

To date, the true prevalence of RAD has not been established and there is a need to know in greater depth which entities are comorbid to this difficulty, as well as the factors associated with it. There is a large gap in the literature on the difficulties associated with RAD. For this reason, the present study aims to determine, through a systematic review, the relationship between RAD and the presence of psychopathology in children and adolescents (objective 1), as well as to determine the existence of differences in the presence of internalising and externalising psychological problems between the group with RAD and groups with other disorders or typical development (objective 2). 

The hypotheses put forward in this review are: 

**Hypothesis** **1.**
*There is a positive relationship between the presence of RAD and greater psychopathology (internalising and externalising problems and other types of difficulties).*


**Hypothesis** **2.**
*On the other hand, this relationship is expected to be stronger in the case of the group with an inhibited RAD-I attachment disorder versus disinhibited DSED.*


**Hypothesis** **3.**
*There is a greater presence of psychological problems and comorbid disorders in children with RAD compared to children and adolescents with ASD, ADHD, social risk and typical development.*


## 2. Materials and Methods

### 2.1. Search Strategy

A systematic review of the scientific literature related to RAD was conducted according to the guidelines established by the PRISMA statement [22]. The search was carried out in October 2022 in the Web of Science, PubMed and Scopus databases. Based on the PICO strategy [23], an attempt was made to answer the following question: is there a greater affectation in children and adolescents diagnosed with RAD compared to children without protective measures or with other types of disorders? 

The final search combined the proposed key elements. The following Boolean (using MeSH terms) expression was therefore used in Web of Science, PubMed and Scopus: ((reactive AND attachment AND disorder) AND (behavior* OR psychological) AND (health OR psychological) AND well-being) in WoS; in PubMED ((reactive attachment disorder) AND (behavior* OR psychological health OR psychological wellbeing)), and in Scopus ((reactive AND attachment AND disorder) AND (behavior* OR psychological) AND (health OR psychological) AND well-being)). This resulted in 770 articles, of which 309 were extracted from the Web of Science, 271 from PubMed and 190 from Scopus. All records were downloaded in an Excel spreadsheet including authors’ names, titles, journals and abstracts. 

### 2.2. Eligibility Criteria

Inclusion criteria were established as documentary typology articles published between 2010 and 2022 in Spanish or English, in which an empirical study was carried out that included at least one group with RAD and that studied the relationship between RAD and psychological problems, as well as the existence of differences between groups with and without RAD, or with other disorders, in children or adolescents. We established a temporal criterion from 2010 to 2022 to examine the literature both preceding and following the publication of the DSM-5. This edition, released in 2013, brought about substantial changes in the conceptualization of RAD.

Exclusion criteria included other types of documents such as conference proceedings, books, book chapters, or grey literature, articles published in languages other than those established in the inclusion criteria, papers that did not include a group with RAD in the study, descriptive articles, narrative reviews, systematic reviews or case studies, and studies that included adults in their samples.

### 2.3. Procedure

All records were imported into Covidence [24], a screening and extraction tool for systematic reviews. Duplicate articles were removed and then screened in a blinded fashion by two authors. When there was disagreement, a third reviewer interceded. 

The number of articles analysed was 770, of which 74 were eliminated as duplicates and 635 were eliminated after reading the title and abstract. After analysis of the full body of the remaining 61 articles, a total of 36 articles were removed for different reasons: focusing on attachment styles but not on RAD (10 articles), being systematic reviews (2 articles), not being experimental but descriptive studies (8 articles), not addressing the consequences of RAD on psychological health (11 articles) and analysing the relationship between RAD and brain structures (5 articles). Following this process, a total of 25 articles were included in the final study (see Figure 1). Subsequently, a narrative summary was used as the method for synthesising the studies, taking into account that the heterogeneity between studies was too great for meta-analysis.

This systematic review was registered in PROSPERO with code CRD42022359220. 

### 2.4. Methodological Quality of the Selected Articles

The quality assessment of the selected articles was carried out by two independent and blinded reviewers. For this purpose, the QUIRE Guidelines 2.0 quality scale [25] was used, selecting 11 of the 18 indicators included in the scale—title, abstract, problem description, available knowledge, specific aims, measures, analysis, results, discussion, interpretation, limitations and conclusions. After its application, two researchers classified the papers into three categories: good quality—when 9 or more of the indicators were met, medium quality—when between 8 and 6 indicators were met, and low quality—when 5 or fewer indicators were met. All papers included in the present review were classified by both researchers as being in the good quality category. The inter-judge agreement following Cohen’s Kappa [26] was 1, a very satisfactory value according to [27].

## 3. Results

Table 1 summarises the articles included in this review, including the authors, year of publication, diagnostic instruments and criteria, main results and limitations of the studies. 

### 3.1. RAD Diagnostic and Assessment Tests Used in the Articles Included in the Systematic Review

The selected articles used nine instruments to assess and diagnose RAD and some adaptations of these. The most commonly used was the Disturbance Attachment Interview—DAI [28]—others that have been used are The Child and Adolescent Psychiatric Assessment—CAPA [29], Relationship Problems Questionnaire—RPQ [30], Reactive Attachment Disorder and Disinhibited Social Engagement Disorder Assessment—RADA [31], Development and Wellbeing Assessment–RAD/DSED Section—DAWBA-RAD/DSED [30], Relationship Patterns Questionnaire—RPQ [32], The Rating of Inhibited Attachment Behavior Scale—RInAB [33] and FinAdo-RAD [34].

Also, some research, refs. [17,35,36,37], did not use assessment instruments as the persons included in the study samples had been previously diagnosed based on the DSM-IV-TR, DSM-5, or World Health Organisation ICD-10 criteria [38], for the most part. A more detailed analysis of this information can be found in Table 1.

### 3.2. Conditions and Previous History of the Sample According to the Articles Included in the Review

The majority of studies (84%; *n* = 21) use children whose life history is characterised by protective measures, such as residential care, foster care or adoption, either because they have experienced or there are well-founded suspicions of maltreatment, neglect or abuse [13,14,15,16,17,31,34,37,38,39,40,41,42,43,44,45,46,47,48,49,50,51]. 

One study used a clinical sample with externalising problems and a general population [16], another used a sample with sociocultural deprivation and suspected malingering [52], two studies used children in special schools with emotional or behavioural problems [53,54], one study used people with intellectual disabilities [35] and one study used children with a diagnosis of RAD but no information on their life history [36]. 

### 3.3. Association between RAD Diagnosis and Other Personal, Social and Mental Health Difficulties

The 25 articles included in this review mainly address the presence of internalising, externalising and social difficulties in children with RAD (Table 1). More specifically, the presence of RAD in children aged 12–62 months in foster care is related to more internalising and externalising difficulties (rho = 0.33, *p* = 0.02) [40]. Also, in general, RAD-I (r = 0.33) and DSED (r = 0.28) are related to more emotional, behavioural and social problems. In this sense, children with RAD-I or DSED showed more symptoms than children who had neither of these difficulties. Those with both RAD and DSED also showed more symptoms [34], with the RAD-I group having higher mean scores than the DSED group for internalising, externalising and total problems.

In the same line, ref. [51] report a significant association between RAD and emotional, social, behavioural and hyperactive difficulties, with coefficients between r = 0.57 (*p* < 0.001) and r = 0.30 (*p* < 0.05). This conclusion was also reached by [54], who related DSED to difficulties in these areas (r = 0.76, *p* < 0.001). Furthermore, he concludes that there are also more conflictual relationships with teachers (r = 0.25, *p* < 0.05) and greater dependence on (r = 0.40, *p* < 0.01) and more trust towards significant others (r = 0.35, *p* < 0.01) and a better global self-concept (r = 0.22, *p* < 0.05) in those with DSED.

In another study, by [53], when the informants are caregivers, RAD-I was positively related to emotional problems (r = 0.41, *p* < 0.01), behavioural problems (r = 0.52, *p* < 0.01; r = 0.32, *p* < 0.01) and problems with peers (r = 0.24, *p* < 0.01; r = 0.28, *p* < 0.05) and negatively related to prosocial behaviour (r = −0.50; r = −0.24, *p* < 0.05). The DSED was associated with conduct problems (r = 0.18, *p* < 0.05), hyperactivity (r = 0.21, *p* < 0.05) and peer problems (r = 0.24, *p* < 0.01). When the assessments are made by teachers, there are positive and significant relationships between the RAD-I group and the SDQ factors, with coefficients between 0.48 and 0.33 (*p* < 0.01) and a negative relationship with prosocial behaviour (r = 0.37, *p* < 0.01). In contrast, for the DSED type, there were no significant relationships.

On the other hand, RAD-I and DSED were not associated with aggressive behaviour. However, their association with lower behavioural inhibition (r = −0.28, *p* < 0.05) and higher ADHD symptomatology (r = 0.45; *p* < 0.001), oppositional defiant disorder, dissocial behaviour disorder (r = 0.30, *p* < 0.01) and depressive disorder was seen over time, at 30 months (r = 0.35, *p* < 0.001), at 42 months (r = 0.72, *p* < 0.001) and at 54 months (r = 0.62, *p* < 0.05). However, in the RAD-I group there was greater functional impairment in different contexts and lower socioemotional competence than in the DSED group [13]. Along the same lines, in a study comparing children who have been adopted with children who have not been adopted, it was observed that 60% of adopted children had RAD, while among non-adopted children there were no cases of RAD. Furthermore, 85% showed comorbid ADHD, 85% had a conduct disorder, 75% had oppositional defiant disorder, 70% had ASD and 55% had PTSD, with these data being significantly higher in children who had been adopted compared to those who had not (*p* < 0.001) [44].

Meanwhile, in the work of [52] it was noted that 75% of children in the RAD group had a clinical range of problems compared to 15% of the general school population, according to the SDQ. Similarly, 52% of children with RAD also had ADHD, 29% had oppositional defiant disorder, 29% had a possible conduct disorder, 14% had ASD and 19% had PTSD. 

The study by [15] obtained similar findings, associating RAD and DSED with internalising (β = 0.832, *p* ≤ 0.001; β = 0.909, *p* ≤ 0.001) and externalising problems (β = 1.96, *p* ≤ 0.001; β = 1.18, *p* ≤ 0.001). Similarly, superficial social relationships were also related to more internalising and externalising problems (β = 1.17, *p* ≤ 0.001) and (β = 2.33, *p* ≤ 0.001), respectively. The same was true for the feeling of unpredictability, which was associated with internalising problems (β = 1.52, *p* ≤ 0.001) and externalising problems (β = 2.70, *p* ≤ 0.001).

However, when different informants are used in the studies, disparate results are obtained, for example, in a sample of adopted children (65% with mental health problems), children with externalising problems and low-risk children (19% with mental health problems), when DSED was reported by teachers it was associated with a conduct disorder (OR = 1.630, *p* < 0.01). In the case of researcher assessment, it was associated with ADHD (OR = 1.95, *p* < 0.01), and in the case of parents with emotional problems (OR = 1.644, *p* < 0.01), with ADHD (OR = 1.792, *p* < 0.01) and Oppositional Defiance Disorder/Conduct Disorder (OR = 1.711, *p* < 0.01) [16]. Similarly, with caregiver assessments, in the case of RAD, it was related to behavioural and emotional problems (rs = 0.60, *p* < 0.001), but this association was not significant with the DSED (rs = 0.30, *p* = 0.118). In the case of teachers as informants, there was a significant relationship between RAD and conduct problems (rs = 0.54) and hyperactivity (rs = 0.46). The same conclusions were reached in the work of [14], where RAD was associated with conduct problems (r = 0.79), hyperactivity (r = 0.77), emotional problems (r = 0.50) and problems with peers (r = 0.62), which was not the case with the DSED.

Similarly, it was observed that in those children with comorbid DSED and RAD, the majority had behavioural problems (73%) and emotional insensitivity (100%) compared to the DSED group (χ^2^ = 8.1, *p* = 0.008). There were no differences between the RAD + DSED group and the DSED group in terms of challenging behaviour, ASD, ADHD-C, ADHD-I, anxiety and depression [45]. On the other hand, a higher incidence of social problems was observed in children with RAD. Thus, children with RAD vs. children without RAD typically trust primary attachment figures such as teachers less (r = −0.28, *p* < 0.05), irrespective of age, the presence of autism or ADHD [39].

Similarly, RAD predicts low social functioning (β = −0.36, *p* < 0.01, 95%, [CI] [−0.33, −0.09]) and low social competence (β = −0.38, *p* < 0.01, 95% CI [−0.05, −0.01]) regardless of the length of time institutionalised and the number of changes of protective measures. On the other hand, DSED indicators (β = −0.38, *p* < 0.001, 95% CI [−0.49, −0.16]), together with the number of changes of measures (β = −0.22, *p* < 0.05, 95% CI [−0.29, −0.01]), predicted worse social functioning. Furthermore, DSED predicted relational victimization (β = 0.29, *p* < 0.05, 95% CI [0.02, 0.14]) and lower social competence (β = −0.29, *p* < 0.01, 95% CI [−0.06, −0.01]) [42]. It has been observed that children with RAD are more likely to be victims of bullying (RR 2.68, 95% CI 1.50–4.77; *p* < 0.001), in particular, these children reported three times more bullying experiences than children without RAD. Similarly, the variable of being bullied (RR 2.08, 95% CI 1.17–3.69, *p* = 0.01) showed 2.8 times higher rates of incidence than in children without RAD [47].

TAR-I symptomatology was significantly associated with lower social acceptance (B = −0.051, *p* = 0.013), athletic competence (B = −0.048, *p* = 0.028), romantic appeal (B = −0.053, *p* = 0.007) and close friendship (B = −0.052, *p* = 0.005), while higher DSED symptomatology was associated with lower scores on scholastic competence (B = −0.125, *p* = 0.038) [50].

### 3.4. Differences between the RAD Group and Other Children with Neurodevelopmental or Typical Developmental Problems

Fourteen articles included comparisons of children with RAD to other children with neurodevelopmental difficulties, typically developing children or children at risk of social exclusion.

Firstly, when comparing children with RAD to children without RAD, it was observed that the former had a more negative internal model of themselves and others [51]. In general, children under protective measures, versus children who were socially at risk, showed more symptoms of RAD (*p* < 0.001) [15]. Ref. [50] finds that the RAD group presented lower intellectual ability (F(1, 39) = 8.78, *p* < 0.01), more hyperactive (F(1, 38) = 10.86, *p* < 0.01), more depressive (F(1, 38) = 9.86, *p* < 0.01), anxious (F(1, 39) = 6.14, *p* < 0.05), depressive (F(1, 39) = 8.80, *p* < 0.01), anger (F(1, 39) = 8.93, *p* < 0.01), post-traumatic stress (F(1, 39) = 10.53, *p* < 0.01), dissociation (F(1, 39) = 11.34, *p* < 0.01) and autism-related symptoms (F(1, 37) = 7.28, *p* < 0.05), than the non-RAD group.

When comparing children with DSED to children without attachment disorders, the former showed higher scores in indiscriminate friendliness (t(35.78) = 9.47, *p* < 0.001), in reliability trust (t(64) = 2.58, *p* = 0.02), in general problem behaviour (t(64) = 8.09, *p* < 0.001), higher dependence on the teacher figure (t(58.26) = 3.76, *p* < 0.001), better overall self-concept (t(64) = 2.47, *p* = 0.02) and higher conflictual relationships with teachers (t(57.95) = 2.90, *p* = 0.005) [54]. Likewise, when adolescents in residential care with and without DSED were compared, the latter had lower mean scores in terms of social acceptance (*p* = 0.012) and self-worth (*p* = 0.037). In the case of adolescents with RAD-I, they presented lower school self-concept (*p* = 0.020) and higher social self-concept (*p* = 0.021) than the control group. On the other hand, the DSED group presented lower mean scores in school self-esteem (*p* = 0.005), social acceptance (*p* = 0.015), athletic competence (*p* = 0.038), physical appearance (*p* = 0.048) and self-worth (*p* = 0.013) than the control group [48].

On the other hand, it was observed that children with RAD-I/DSED or ADHD have more behavioural problems and hyperactive symptoms than children with ASD; differences that were found to be statistically significant (*p* < 0.001). When comparing the RAD-I/DSED group with children with ASD, the RAD-I/DSED group showed fewer peer problems (*p* < 0.001), more behavioural problems (*p* = 0.008) and more prosocial behaviour (*p* = 0.006). However, children with ADHD showed more hyperactive symptoms than children with RAD-I/DSED (*p* < 0.001) [17]. 

In contrast to these results, the study by [45] showed that children with RAD and/or DSED showed higher percentages of more oppositional behaviour, conduct problems, impulsivity, hyperactivity and depression than the group of children with ADHD-I, as well as a higher percentage of conduct problems than the group of children with ADHD-C and the group of children with ASD. These differences in the percentages of problems between groups were statistically significant (χ^2^ = 6.8–128.8, *p* = 0.009–*p* < 0.001). Also, children with RAD and/or DSED showed more lying, stealing, unresponsive and overeating behaviour than children with ADHD-I, ADHD- or ASD. Likewise, children with RAD vs. children with ASD and typically developing children report more social difficulties in the use of context (t(81) = 2.886, *p* = 0.005), rapport (t(83) = 4.173, *p* < 0.001) and social relationships (t(82) = 2.849, *p* = 0.006). It was observed that up to 60% of these children meet the diagnostic criteria for autism [37].

In addition, children with RAD showed more socially disinhibited behaviour, hypervigilance and greater comfort with strangers than children with ASD (*p* < 0.05). In addition, children with RAD were more likely than children with ASD to report anxiety disorders (73% vs. 18%), ADHD (49% vs. 29%) and conduct problems (27% vs. 2%) 42. However, children with RAD were also found to be more likely to have developmental speech and language disorders, ASD, ADHD, intellectual disability, anxiety disorders, conduct disorders, stress-related disorders, tic disorders and affective disorders than children with neurotypical development [36].

The comorbidity of RAD with DSED significantly increased the likelihood of showing more internalising problems, externalising problems and difficulties in general (*p* < 0.001) than children without personal difficulties or children with RAD or DSED alone [34]. When assessed in a sample of children with intellectual disabilities with and without attachment difficulties, it was observed that those with RAD showed more difficulties with daily living skills (*p* = 0.017) and communication (*p* = 0.037) than children without RAD. Similarly, children with DSED showed more problems with emotional (*p* = 0.023), socialisation (*p* = 0.044) and motor skills (*p* = 0.016). In both cases, children with RAD (*p* = 0.023) and children with DSED (*p* = 0.003) showed more egocentric behaviour than children without attachment problems, and more antisocial behaviour problems (*p* = 0.047) [35].

Along the same lines, ref. [43] focused their study on children in foster care. They differentiated the children into four groups: DSED, RAD-I, DSED+RAD-I (comorbid group) and no attachment disorder. Their results indicate the existence of differences between groups concerning the presence of psychopathology (F(18, 267) = 2.15, *p* = 0.005). In particular, parents reported that the DSED group had more clinical problems than the group without symptoms (F(1, 101) = 3.97–8.80, *p* = 0.049–0.004). Related to the above, when children presented DSED and RAD (comorbid), they show more difficulties in general (F(12, 178) = 3.00, *p* = 0.001) but also more internalising (X = 76.84; SD = 17.04, *p* < 0.001) and externalising (X = 73.24; SD = 12.45, *p* < 0.001) problems than children without difficulties. On the other hand, children with DSED showed more internalising (X = 57.48; SD = 13.66, *p* < 0.05), externalising (X = 61.14; SD = 13.75, *p* < 0.01) and general difficulties (X = 60.35; SD = 13.68, *p* < 0.01) than children without difficulties. The same conclusions were reached by teachers when they assessed the group of children with RAD and DSED. They found more internalising problems (X = 61.05; SD = 16.97, *p* < 0.01), externalising problems (X = 66.83; SD = 12.91, *p* < 0.05) and total problems (X = 60.60; SD = 10.53, *p* < 0.05) than the group without attachment-related difficulties. These results are similar to those found by [46], who found that the group of children with RAD had higher percentages of emotional problems (60% vs. 36%), conduct problems (100% vs. 71%), hyperactivity (67% vs. 21%) and problems with peers (87% vs. 71%) than children without attachment problems.

**Table 1 children-10-01892-t001:** Studies linking the presence of RAD to psychological problems and group differences.

Article	Country of Sample and Life History or Protection Measures	Sample	Instrument Used to Assess RAD or DSED or Diagnostic Criteria Used	Main Results	Study Limitations
Bosmans et al. (2019) [39]	Belgium. Children in special education schools. A total of 48% suspected to suffer from mistreatment, abuse or neglect.	Children with RAD: *n* = 21. Children without RAD: *n* = 46. Mage: 8.70 years, SD = 0.99. In total, 27.2% were diagnosed or suspected with RAD.	Disturbance Attachment Interview—DAI [28].	RAD correlates negatively with trust in teachers and emotional security when compared to the group without RAD.	They only focus on children’s emotional safety in relation to teachers, rather than on their general relationship.
Bruce et al. (2019) [40]	Scotland. Children in foster care.	Moment 1: *n* = 100. Moment 2: *n* = 76. Age: 12–61 months.	The Rating of Inhibited Attachment Behavior Scale—RInAB [33].	There is a relationship between RAD symptomatology and the presence of internalising and externalising problems, as well as lower verbal and total IQ.	Small sample size and small number of children with RAD. There was a change in protective measures between moment 1 and moment 2. The procedure used to activate the attachment mechanism in children may not be sufficiently stressful. There was a Change of caregivers at moment 1 for some children.
Coughlan et al. (2021) [17]	England. There is no information on the children’s previous history.	Group RAD (*n* = 39). Group ADHD (*n* = 1.430). Group ASD (*n* = 1.193). Age < 17 years.	World Health Organisation ICD-10 [38].	There are significant differences between groups (*p* < 0.001) in all factors of the SDQ questionnaire. The ASD group had more emotional problems, difficulties with peers and fewer prosocial strengths than the ADHD or RAD group. ADHD and RAD groups had more hyperactivity and behavioural problems.	Additional information from the children’s context is not available for a better understanding of the results.
Davidson et al. (2015) [41]	Scotland RAD group had a history of abuse and fostering.	Children 5–11 years. Group RAD: *n* = 67. Group ASD: *n* = 58.	Disturbance Attachment Interview—DAI [28].	Higher prevalence of comorbidity with other mental health problems, as well as more disinhibition and indiscriminate friendliness in the RAD group.	Failure to discriminate between different types of RAD. There is discrepancy between the RAD diagnostic criteria (DSM-IV) and the RAD assessment instrument, which follows DSM-5 criteria.
Elovainio et al. (2015) [34]	Finland. Adopted children.	*N* = 853. Age: 6–15 years. No RAD *n* = 348; DSED *n* = 153; RAD *n* = 137; comorbid (DSED + TAR) *n* = 214.	Disturbance Attachment Interview—DAI [28].	Relationship between RAD and DSED with the presence of emotional, hyperactive and behavioural symptoms. The comorbid group had greater internalising, externalising and total problems than the other groups.	Does not allow for causal relationships between attachment disorders and psychological problems. Heteroinformed assessment (adoptive parents).
Giltaij et al. (2016) [35]	The Netherlands. No data on life history. RAD group had higher indicators of neglectful care.	*N* = 55. Age: 5–11 years. Mild intellectual disability. RAD (*n* = 3); DSED (*n* = 1); RAD + DSED (*n* = 6). Control group (*n* = 45).	Diagnosis was based on the criteria of the DSM-5 [6].	RAD + DSED group has greater difficulties in adaptive behaviour in the area of socialisation and motor skills. RAD and/or DSED group had more disruptive and antisocial behaviour than the control group. DSED had greater emotional problems compared to the non-DSED group.	Hetero-information (teachers).
Gleason et al. (2011) [13]	Romania Children who had spent an average of 86% of their lives in institutional care.	*N* = 136 initially (6–30 months). These were followed up at 30, 42 and 54 months of age. *n* = 68 assigned to care as usual. *n* = 68 in foster care.	Disturbance Attachment Interview—DAI [28].	DSED criteria met at study start 41/129 (31.8%), 30 months 22/122 (17.9%), 42 months 22/122 (18.0%) and 54 months of age 22/125 (17.6%). RAD-I criteria met at study entry 6/130 (4.6%), 30 months 4/123 (3.3%), 42 months 2/125 (1.6%) and 54 months of age 5/122 (4.1%).	Neglectful care conditions other than institutionalisation are not assessed. The life history and background of caregivers (maltreatment or mental health status) are not considered. Low rates of emotionally withdrawn/inhibited RAD.
Guyon-Harris et al. (2019) [42]	Romania. Foster care.	Experimental group (*n* = 55). Control group (*n* = 50). Mage = 12.80, (SD = 0.71) years.	Version of DAI-EA [20] adapted from the Disturbance Attachment Interview—DAI [28].	A greater presence of RAD signs predicts worse overall social functioning and lower social competence. A greater presence of DSED signs predicts worse social functioning, more relationship victimisation and lower social competence.	Adolescent self-reports are not included. Signs of RAD are examined, but not the diagnosis. Focuses on social functioning and does not explore other domains.
Hong et al. (2018) [36]	Corea del Sur. General population.	Children < 10 years. *N* = 14,029,571; RAD = 736.	Diagnosis was based on the criteria of ICD-10 [38].	Higher prevalence of comorbid disorders in children with RAD than in typically developing children. This type of disorder varies according to age group.	Patients who underwent treatment outside the hospital are not included. The precise incidence of RAD is not recorded.
Jonkman et al. (2014) [43]	Netherlands Children who had experienced at least one foster care disruption.	*N* = 126 in foster care families. Age: 24–72 months (with the foster family). Control group *n* = 84; RAD-I: *n* = 11; DSED: *n* = 19; RAD-I + DSED *n* = 12.	Disturbance Attachment Interview—DAI [28].	RAD and DSED groups presented greater internalising, externalising and total problems.	Small sample size. Observational measures. Hetero-informed assessment (family members and teachers).
Kay y Green (2013) [15]	England Residential care and children at risk of social exclusion.	RAD foster care group: *N* = 153. RAD social risk group: *n* = 42. Age: 10–15 years.	Development and Wellbeing Assessment–RAD/DSED Section (DAWBA-RAD/DSED; [30] 24-item version).	Greater presence of RAD symptomatology in the group in foster care compared to the social risk group. There was a relationship between RAD symptomatology and greater behavioural and emotional problems and a lower adaptive capacity.	Hetero-information (social workers). Incomplete data on participants’ life history.
Kay et al. (2016) [16]	England Adoption. Without prior institutionalisation.	Adoption group: *n* = 60. Group with externalising disorder: *n* = 26. Group at social risk: *n* = 55. Age: 6–11 months.	CAPA-RAD [16,41] adaption of CAPA [29] based on the DSM-5 criteria [6].	In the adoption group, there was a significant relationship between the presence of DSED and ADHD according to the teachers. According to parents, this relationship was found with ADHD, emotional and behavioural problems.	Small sample size. Biases in sample selection. No clinical diagnosis of DSED. Lack of pre-adoption information on children’s life history. Hetero-informed assessment (family members and teachers).
Kocovska et al. (2012) [44]	United Kingdom. Adopted children who had suffered abuse, neglect, abandonment and abuse.	*N* = 66. Age: 5–12 years. Adopted group *n* = 33 of which 20 had RAD. Control group *n* = 32.	Relationship Patterns Questionnaire (RPQ) [32].	The group of adopted children presented greater behavioural problems than the group of non-adopted children. High prevalence of children with RAD in the adopted group.	Possible bias in the sample of adopted families, as those with higher motivation and better family functioning participated. The attachment assessment measure used has shown adequate psychometric properties in children under 8 years of age and the sample is somewhat older.
Lehmann et al. (2016) [14]	Norway Foster care.	*N* = 122. Age: 6–10 years.	Reactive Attachment Disorder and Disinhibited Social Engagement Disorder Assessment—RADA [31].	The positive association between RAD symptoms and DSED. RAD is associated with more comorbid personal difficulties than DSED.	Small sample size. Hetero-informed (caregiver) assessment. Broad, general categories of experiences of maltreatment.
Mayes et al. (2017) [45]	United States. Children who had suffered severe abuse or neglect (*N* = 16) and children adopted from Russian or Chinese orphanages (*N* = 4).	RAD + DSED *n* = 15; DSED *n* = 5; ASD *n* = 933; ADHD-C *n* = 631; ADHD-I *n* = 264.	The Rating of Inhibited Attachment Behavior Scale—RInAB [33].	RAD group: all of them had callous–unemotional traits and 73% had behavioural problems. RAD-I+DSED group: more behavioural problems and emotional insensitivity than the DSED group. The RAD group had more behavioural and emotional problems than the group with ASD, ADHD-C and ADHD-I.	Unrepresentative sample of children with RAD because of their size and because they initially presented other personal difficulties that required referral.
Moran et al. (2017) [46]	Young people in the Youth Justice Service. In total, 86% had experienced abuse.	*N* = 29. Age: 12–17 years. Mage = 16.2. SD = 1.3. Carers: *n* = 29. Teachers: *n* = 20.	The Child and Adolescent Psychiatric Assessment—CAPA [29].	Relationship between RAD symptoms and other mental health problems. More mental health problems in the RAD group than in the non-RAD group.	Does not allow causal relationships to be established. Hetero-assessment (parents and teachers). It is not known whether the children had any other psychological or neuropsychological problems.
Pritchett et al. (2013) [52]	England. A general population with socio-cultural deprivation. RAD group had histories of abuse.	*N* = 1600. Age: 6–12 years. RAD group *n* = 22. Of these 13 had a diagnosis of RAD and 9 had suspected RAD.	The Child and Adolescent Psychiatric Assessment—CAPA [29].	RAD group had a high comorbidity with other disorders and presence of behavioural problems.	Small sample size.
Raaska et al. (2012) [47]	Finland. Children adopted between 1985 and 2007 from different countries.	*N* = 365. Age: 9–15 years. (47.8% men.)	FinAdo-RAD [33].	There is a significant relationship between the severity of RAD symptomatology and participation in bullying situations, both victimisation and peer bullying.	Relatively low response rate. Children completed the questionnaires at home (possible parental influence). There is no information on whether RAD also relates to those children who engage in bullying in normative samples.
Sadiq et al. (2012) [37]	Muestra con TAR: 1/3 vivía con la familia biológica y 2/3 eran adoptados.	*N* = 126. Age: 5–8 years. RAD group: *n* = 35. ASD group: *n* = 52. TD group: *n* = 13.	Diagnosis is based on the criteria of World Health Organisation ICD-10 [38].	The RAD group had more difficulties in the use of context, rapport and social relations than the ASD group.	Clinical sample and not general population.
Seim et al. (2021) [48]	Norway Residential care. In total, 71% exposed to abuse.	*N* = 306. Age: 12.2–20.2 years. Of these, children with RAD *n* = 28; children with DSED *n* = 26. Control group: *n* = 10,480. Age: 12–20 years.	The Preschool Age Psychiatric Assessment—PAPA [55]—adaptation of CAPA [29] for children between 2 and 8 years.	The RAD group had lower self-esteem in school competence and higher self-esteem in close friendships than the control group. The DSED group had lower self-esteem in social acceptance, sports competence, romantic attractiveness and close friendships than the control group.	Hetero-information (caregivers). Inability to determine whether symptoms were present before the age of 5 years.
Seim et al. (2022) [49]	Norway. Foster care. Exposure to neglect. In total, 71% exposed to abuse.	*N* = 381. Age: 12.2–20.2 years. RAD (*n* = 33). DSED (*n* = 31). RAD + DSED (*n* = 2).	The Preschool Age Psychiatric Assessment—PAPA [55]—adaptation of CAPA [29] for children between 2 and 8 years.	High prevalence of mental disorders in children with RAD and DSED. Children with RAD and/or DSED had a high prevalence of emotional and behavioural problems.	Hetero-assessment (caregivers). Difficulty in demonstrating the presence of RAD before the age of 5 years.
Shimada et al. (2015) [50]	Japan. Residential foster care. RAD Group has suffered abuse and neglect.	RAD group: *n* = 21. Mage = 12.76 years. Group control: *n* = 22. *M*_age_ = 12.95 years.	Diagnosis based on the criteria of DSM-5 criteria [6].	The RAD group had smaller volume of grey matter. This is related to greater internalising and externalising problems according to the SDQ.	Small sample size. Cross-sectional design. Differences in IQ between RAD group and control group.
Vervoort et al. (2013) [53]	Belgium. School-aged children with emotional and behavioural problems. In total, 25% diagnosed or suspected RAD, and 48% suffered maltreatment, abuse or neglect.	Children DAI *n* = 77. Children RPQ *n* = 152. Mage 7.92 years.	Disturbance Attachment Interview—DAI [28]. Relationship Problems Questionnaire—RPQ [30].	More frequent and stronger associations between RAD and emotional and behavioural difficulties than DSED.	Small sample size. Need to use other measures to confirm the diagnosis of RAD.
Vervoort et al. (2014) [54]	Belgium. Special education children with emotional and behavioural problems (suspected DSED).	DSED group: *N* = 33 special education. Control group: *N* = 33 general education. Mage 8.52 years	Relationship Problems Questionnaire—RPQ [30].	DSED group showed more indiscriminate kindness and more behavioural problems than the control group. DSED group showed more positive overall self-concept and greater confidence in relationships than general education children.	The study focuses on one DSED indicator (indiscriminate kindness). Cross-sectional study.
Zimmerman & Iwanski (2019) [51]	Germany. Children at risk of RAD-I had experienced severe neglect or abuse.	*N* = 64 children in institutions and the general population. Age: 5-10 years. Of these, 32 suffered from RAD (foster homes or families).	Relationship Patterns Questionnaire—RPQ [32].	The RAD risk group had a poorer self-concept, a greater number of negative signals from others through Internal Working Models and greater mental health problems. Positive association of RAD with personal difficulties and negative association with prosocial behaviour.	Not specified.

## 4. Discussion

In line with objectives 1 and 2 of this study, the aim was to determine whether RAD (Reactive Attachment Disorder) was associated with other psychological difficulties (objective 1), as well as to assess whether these difficulties were greater than in other groups with psychological or developmental difficulties, or those without difficulties. We observed that the results obtained in the present review point to a relationship between the presence of RAD, inhibited, disinhibited or mixed, and other mental health and developmental problems, such as emotional symptoms (anxiety and depression), dissociative symptoms, stress-related difficulties (PTSD), behavioural problems (oppositional defiant disorder, dissocial disorder), relationships with peers, hyperactivity and autism or intellectual disabilities, among others. In addition to the above, children with RAD-I or DSED also present other difficulties with self-concept, social skills, bullying (victims and aggressors) or prosocial behaviour [13,14,15,16,34,38,39,40,41,42,43,44,45,46,47,50,51,52,53].

Research indicates that these problems are mainly related to the presence of a TAR-I and, to a lesser extent, to the presence of a DSED [14,34,41,45,52], with the presence of personal and psychological difficulties being greater when RAD and DSED coexist [43]. All these data would confirm the first hypothesis of the study. Other qualitative studies, such as the one by [54], support these results by finding that children with RAD have more tantrums, which last longer, with the severity of the tantrums increasing with age. Attacks of rage are mainly directed towards the primary caregiver, suggesting that the child may use the caregiver, with whom they feel secure, as a way to release accumulated tensions due to their inability to cope with the tasks and demands of their enivronment. The authors point to the relationship that may exist between implicit memory and anger attacks, addressing cognitive aspects related to the behaviour of children with RAD [56], an aspect that seems to be related to the neglect they suffered in the first years of life, affecting their capacity for self-regulation [48]. These results underscore the relevance of early interactions and life experiences in the development of RAD. 

These problems, in turn, have repercussions at the social level. Ref. [47] points out that schoolchildren with symptoms of RAD are more likely to engage in victimisation or bullying towards their peers. This could be explained by their difficulty in early bonding with their attachment figures, misinterpreting the behaviours of other children, leading to victimisation or bullying during the school years, and perceiving the behaviours of others as possible threats, even in safe care settings. These results suggest the relevance of implementing programmes in school settings that promote healthy relational environments, as well as strategies for emotional regulation. In this way, both children with RAD and others could benefit from these measures, reducing bullying behaviours within the school environment and providing protection to children with RAD, who may have already experienced adverse situations in childhood, thereby preventing further victimization.

Several authors highlight disinhibition/indiscriminate friendliness as one of the factors most present in children with RAD [15,40,53]. The hyper-sociable behaviours and positive appraisals they make of themselves and others could be related to a possible disconnection from their negative parenting experiences and challenging behaviours, as they play a defensive or protective role for themselves. Importantly, overreliance on others can lead to the construction of unhealthy future relationships [52]. Therefore, the detection and psychological treatment of these children become crucial. This way, it may be possible to prevent subsequent romantic relationships in which inequality becomes the norm.

To the second hypothesis, which posed a greater presence of problems and comorbid disorders in the group with attachment disorders compared to other groups, the results indicate more problems in the RAD and/or DSED group compared to children who do not have disorders. Ref. [39] reports lower trust in teachers and lower emotional security in the RAD group. Ref. [32] identified them as having more internalising and externalising problems, more hyperactivity and more behavioural problems. In ref. [44], more internalising and externalising problems were observed as rated by parents or caregivers, while in the case of teacher ratings, these were only when compared to the mixed group and not to the DSED group. Ref. [36] saw RAD with a higher comorbidity with psychiatric disorders. In ref. [46], children with RAD a higher presence of emotional, behavioural, hyperactivity and peer problems. Ref. [48] reported lower self-concept in different areas, as did [30]. Ref. [50] reported greater problems related to traumatic experiences such as anxiety, depression, anger and dissociation, among others, as well as more internalising and externalising problems, according to parents. Ref. [52] found in the DSED group worse self-concept, more conflictual relationships, greater indiscriminate friendliness, reliability trust and general problem behaviour. Knowing that the presence of multiple mental health issues complicates the treatment and course of these conditions, it is crucial to establish standardized assessment protocols in primary care to detect cases as early as possible.

However, it is necessary to emphasize that [35] found no differences in the presence of comorbid disorders between the RAD group and the groups with ASD and ADHD. Due to publication bias, it is possible that studies demonstrating no increased likelihood of comorbidity between RAD and other mental health problems may not have been published. It is essential to note that an article contrary to the proposed hypothesis was found during the review process. On the other hand, according to parents’ assessments, the RAD-I and DSED groups show a greater presence of disruptive and antisocial behaviour in their children. Teachers reported more emotional problems in the DSED group and communication disorders in the RAD-I group.

When comparing the RAD group with the ASD group, ref. [17] reports greater behavioural problems and hyperactivity in the RAD group, as well as fewer problems with peers and greater prosociality. Ref. [41] reports a higher presence of comorbid disorders such as anxiety disorders, ADHD and conduct problems. Ref. [37] reports an increased presence of conduct and emotional disorders. Ref. [38] reports greater difficulties in the use of pragmatics in language and a high relationship with ASD symptomatology.

Concerning the ADHD and RAD groups, ref. [46] reports a greater presence of conduct problems, oppositional behaviour and depression, while [36] reports a greater presence of anxiety disorders, adjustment disorders, and tics, among others, depending on the age of the children. On the other hand, ref. [17] reports greater hyperactivity in the ADHD group than in the RAD group.

All these results allow us to accept the second hypothesis of the study, which is that the group with RAD presents greater problematicity at different levels than children and adolescents without attachment disorders, with ADHD or with ASD. It would be interesting, in the future, to explore the potential relationship between RAD and other neurodevelopmental challenges.

Despite the contributions of the systematic review work we carried out, it should be noted that the search strategy was not developed by a librarian. On the other hand, the main limitations of this study are the number of databases consulted and the number of articles studied. The three main databases in the field (Web of Science, PubMed and Scopus) have been used, but it would be interesting in the future to consider other databases such as ProQuest Central and Cochrane Library. Similarly, the present review was limited to a time span between 2010 and 2022, so future work could review the literature over a longer time span. Also, we only considered the research in English and Spanish, so it is possible that not all of the existing literature was accessible. On the other hand, it is common for there to be a publication bias where articles that do not obtain positive results are not published. Given this bias, it is possible that the results are overestimated. 

Future studies should mainly address the study of the predictor variables of RAD, controlling for aspects such as the type of protection measures, changes in the measures taken, the quality of care or the typology that leads to a situation of neglect. All of this takes into account children’s relationship with the age variable. Furthermore, it may be important, both from an empirical and practical point of view, to study specific interventions aimed at reducing the emotional, psychological and social problems presented by children with RAD. To this end, it could be considered whether interventions aimed at addressing problems associated with insecure attachments, or specific programmes associated with traumatic situations, may have a beneficial effect in the case of RAD, or whether specific interventions focused on attachment disorders could be developed.

Given the scientific evidence pointing to the consequences on the psychological health of children with RAD, it is worth considering a diagnostic assessment in this sense, confirming, or not, the presence of the disorder, which can help us to understand the behaviours and ways to relate to these children, giving guidance to caregivers, professionals and teachers, which can help, in the specific case of foster care, to assist child–family adaptation and prevent ruptures that lead to the failure of the placement and therefore lead to a change of caregiver; a variable related to the presence of RAD, but also to its maintenance.

Moreover, these results must be taken into account in the design of protection policies, avoiding as much as possible the transit of the child from one placement to another and from one caregiver to another, encouraging the most stable measures possible from the moment the child leaves his or her biological family. It can be concluded that the presence of RAD has negative consequences on the mental health of children and adolescents. These consequences are greater in the case of RAD-I compared to DSED, while they more important in this group of children than in children with other disorders, such as ADHD or ASD.

In conclusion, the synthesis of existing research consistently reveals a compelling association between RAD and an elevated risk for the emergence of additional psychological disorders. This finding emphasizes the imperative of recognizing and addressing RAD in early intervention and therapeutic strategies to mitigate its potential cascading effects on mental health. As we strive for a comprehensive understanding of psychopathology, acknowledging its intricate interplay with RAD opens avenues for targeted interventions and highlights the need for further research to elucidate the nuanced mechanisms underlying this relationship. The implications of this study extend beyond attachment theory, offering valuable insights into the impact of life history on the development of a Reactive Attachment Disorder (RAD), and, consequently, its association with other mental health diagnoses. This study contributes to delineating the probable developmental profiles of the psychological challenges linked to adverse childhood experiences. These findings are indispensable for informing strategies related to the prevention and treatment of children requiring assistance.

## Figures and Tables

**Figure 1 children-10-01892-f001:**
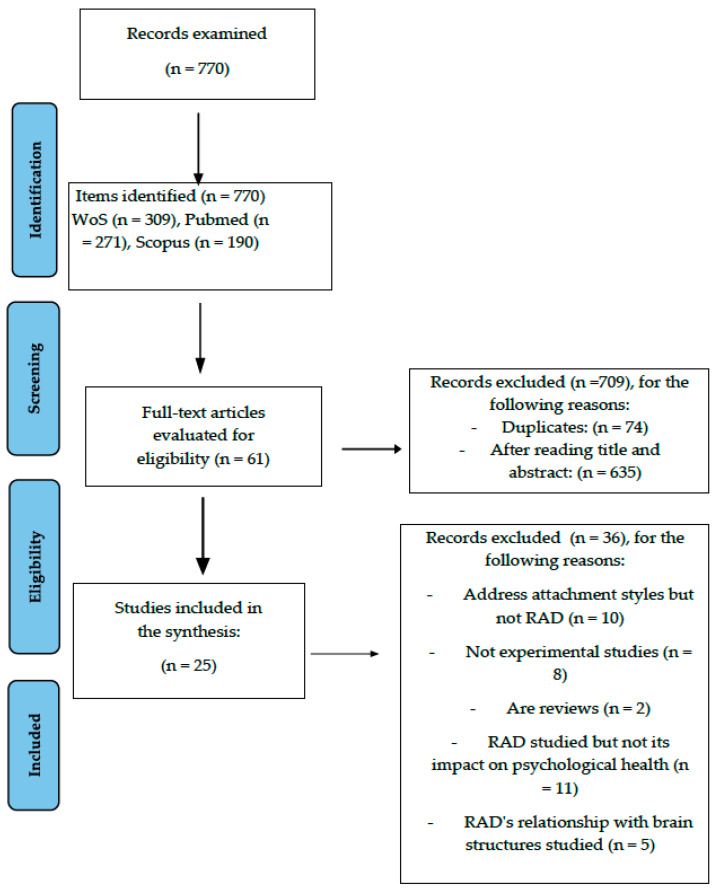
Flow of information through the different phases of a systematic review.

## Data Availability

Not applicable.

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
