# Peer review of "Reactive Attachment Disorder and Its Relationship to Psychopathology: A Systematic Review"

_children, 2023, doi:10.3390/children10121892_

Round 1

Reviewer 1 Report

Comments and Suggestions for Authors The article appears to be well understood, however major revisions appear necessary.
Let me offer some suggestions:
The authors refer to the DSM-IV in the introduction and should explain why they used a manual of disorders that is no longer in use. DSM 5Tr is currently in use. The hypotheses should also be formulated more clearly with respect to what type of "relationship" one expects to find The study analyzed 25 articles. 709 articles were excluded because the authors did not find the full text. This is a questionable and unscientific criterion. The full text can be purchased or requested directly from the authors. Authors should estimate the sample size needed to meet the study hypotheses. There are various calculation techniques to obtain the necessary sample size. There would be only one study with results that conflict with the study's hypotheses. In reality, the results are affected by the high number of studies excluded from the analysis and this limits the scope and extensibility of the results. The limits section needs to be significantly expanded. The selection criterion of the selected articles simply followed an easy-to-find criterion leading to a very small number of articles considered. The investigation carried out remains only on a more superficial level. There is a risk of bias in the confirmation of study hypotheses

Comments on the Quality of English Language he text is clear and easy to understand  

Author Response

Dear Editor and reviewers, thank you for your time and effort dedicated to improving our work. We have incorporated all the suggestions you have made and clarified those aspects you have commented on to make it easier for readers to read.

You will find the answers to your suggestions in this document and your additions in blue in the manuscript.

Thank you again.

Comments and Suggestions for Authors (Reviewer 1)

The article appears to be well understood, however major revisions appear necessary.
Let me offer some suggestions:
The authors refer to the DSM-IV in the introduction and should explain why they used a manual of disorders that is no longer in use. DSM 5Tr is currently in use.

Response: Dear reviewer, considering that our review spans from the year 2010 to 2022, it is relevant to clarify in the introduction the conceptual and diagnostic differences that exist in the various versions of the DSM. Therefore, this aspect is briefly defined, also including information about the DSM-5-TR, the latest available version. Additionally, most studies, even when the DSM-5 and DSM-5-TR were in effect, use assessment instruments based on the DSM-IV or DSM-IV-TR. Thank you for your question (Page 1).

The hypotheses should also be formulated more clearly with respect to what type of "relationship" one expects to find

Response: This issue has been emphasized on page 2.

The study analyzed 25 articles. 709 articles were excluded because the authors did not find the full text. This is a questionable and unscientific criterion. The full text can be purchased or requested directly from the authors. Authors should estimate the sample size needed to meet the study hypotheses. There are various calculation techniques to obtain the necessary sample size.

There would be only one study with results that conflict with the study's hypotheses. In reality, the results are affected by the high number of studies excluded from the analysis and this limits the scope and extensibility of the results. The limits section needs to be significantly expanded. The selection criterion of the selected articles simply followed an easy-to-find criterion leading to a very small number of articles considered. The investigation carried out remains only on a more superficial level. There is a risk of bias in the confirmation of study hypotheses

Response: We apologize for any confusion that may have arisen. As reflected on page 3 (and in Figure 1 page 4):

“The number of articles analysed was 770, of which 74 were eliminated as duplicates and 635 were eliminated after reading the title and abstract. After analysis of the full body of the remaining 61 articles, a total of 36 articles were removed for different reasons: Focusing on attachment styles but not on RAD (10 articles), being systematic reviews (2 articles), not being experimental but descriptive studies (8 articles), not addressing the consequences of RAD on psychological health (11 articles) and analysing the relationship between RAD with brain structures (5 articles).”

All literature meeting the eligibility criteria has been considered. In all cases, we have accessed the full text of the articles if they have progressed to the second round of evaluation. We appreciate your input and hope that with this clarification, the information we have provided in the article (both written and in the form of a figure) is clearer.On the other hand, reasonable and common eligibility criteria have been considered (temporality, document type, study type, languages...): “Articles published between 2010 and 2022 in Spanish or English, conducting an empirical study that included at least one group with Reactive Attachment Disorder (RAD) and examined the relationship between RAD and psychological problems. The study also explored differences between groups with and without RAD or with other disorders in children or adolescents. Exclusion criteria comprised other document types such as conference proceedings, books, book chapters, or grey literature, articles published in languages other than those established in the inclusion criteria, papers lacking a group with RAD, descriptive articles, narrative reviews, systematic reviews, or case studies. Studies including adults in their samples were also excluded.”

All articles meeting these criteria were included in the final synthesis, resulting in the analysis of 25 scientific studies. In no case was the 'accessibility' of the article a selection criterion. We hope that this clarifies the information provided on pages 3 and 4. We want to once again express our gratitude to the reviewer for their effort and dedication to our work

Reviewer 2 Report

Comments and Suggestions for Authors

Thank you for the opportunity to review this paper. 

- According to the methodology, the hypothesis should be stated in the present tense: "There is a relationship..."

- why those databases were chosen? The Authors consider it a limitation but do not explain why they decided to analyse those databases.

- what was the reason for considering 2010 as a starting point?

- 3.1 "Research has used nine instruments" one paper only? 

Author Response

Dear Editor and reviewers, thank you for your time and effort dedicated to improving our work. We have incorporated all the suggestions you have made and clarified those aspects you have commented on to make it easier for readers to read.

You will find the answers to your suggestions in this document and your additions in blue in the manuscript.

Thank you again.

Thank you for the opportunity to review this paper. 

- According to the methodology, the hypothesis should be stated in the present tense: "There is a relationship..."

Response: This aspect has been modified (Page 2).

- why those databases were chosen? The Authors consider it a limitation but do not explain why they decided to analyse those databases.

Response: This information has been included in the discussion (page 17). We have selected these bases as being among the most relevant to the field of mental health. However, the importance of including other databases in future work is also recognised in this section (Page 17).

- what was the reason for considering 2010 as a starting point?

Response: Information has been included on page 3. In particular, our aim was to integrate information from before and after the publication of the DSM-5, which we know had a major impact on this diagnosis.

- 3.1 "Research has used nine instruments" one paper only? 

Response: The wording has been amended. Thank you for your input “The selected articles used nine instruments to assess and diagnose RAD and some adapta-tions of these”.

We would again like to thank the editor and reviewers for their contributions.

Reviewer 3 Report

Comments and Suggestions for Authors

I would like to thank the editors and authors for the opportunity to review the article "Reactive attachment disorder and its relationship to psychopathology: A scoping review"

Treatment approaches aim to repair and strengthen the attachment bond between the child and caregiver to mitigate the impact of early trauma on the individual's psychological well-being.

The chosen theme is interesting and of greater importance.

In this sense, it is important to map the scientific evidence on the relationship between Reactive Attachment Disorder (RAD) and the presence of psychopathology in children and adolescents and determine the existence of differences in the presence of internalizing and externalizing psychological problems between the RAD group and groups with other disorders or with typical development.

In general terms, I can say that the article presents specific references in interest, however most of them in the introduction and justification of the study are more than 5 years old (more than 77%).

The chosen theme is interesting and of greater importance.

I will now offer my contributions or suggestions for improving the manuscript:

Title/ Abstract

It does not identify a systematic review in the title.

It is not clear whether the authors want to carry out a scoping review (as mentioned in the title), or a systematic review (as mentioned in the abstract). I think it will be an oversight in the title description. Authors must review the title or method referred to in the text.

1. Introduction

It is suggested that authors make an explicit statement of all objective(s) or question(s) the review addresses, expressed in terms of a relevant question formulation framework.

This situation must be clear both in the summary and in the Introduction.

2. Materials and Methods

2.1. Search strategy

Sugere-se integrar no manuscrito um quadro de População, Intervenção, Comparador, Resultado (PICO) ou uma de suas variantes, para indicar as comparações que serão feitas.

It is advisable to review the equations boolean, as some seem to lack parentheses in the Boolean conjugation when "OR" is used.

For example, Where is it described,

"...in Scopus (reactive AND attachment AND disorder) AND (behavior* OR psychological AND health OR psychological AND well-being)

I think the authors wanted the following:

"...in Scopus ((reactive AND attachment AND disorder) AND (behavior* OR psychological) AND (health OR psychological) AND well-being))

I would ask the authors to confirm and to see the following Boolean equations.

I miss the authors, in point 2.3. Procedure, indicate which the method for synthesizing the studies: for example, whether it was meta-analysis or narrative summary for quantitative data or meta-aggregation/meta-synthesis for qualitative data.

3. Results

In point 3.3. Association between RAD diagnosis and other personal, social and mental health difficulties.

The authors describe "The 18 articles included in this review mainly address the presence of internalising, externalising and social difficulties in children with RAD (Table 1)."

It seems to me that the authors wanted to refer to 25 articles and not 18. I ask to confirm.

4. Discussion

The discussion reads like a general summary of the results.

I suggest that the authors present the results but contextualize them with the existing literature, demonstrating the importance and contribution of the topic and results to clinical practice and new investigations.

The authors discuss the limitations of the studies found. However, they need to discuss the limitations of the systematic review process, for example, they opted for articles only in Spanish and English.

Conclusion

I miss a chapter with a conclusion in which the authors present the main conclusions in response to the question of research, and value the contributions of the work in relation to the area in question.

I think that at the end of the discussion they form part of the conclusion of the work, I would suggest moving this last paragraph of the discussion to the conclusion.

References

Some of the references should be reviewed as some do not comply with the journal's standards.

Most references are more than 5 years old (more than 77%). It is suggested that more recent references be included in the introduction and discussion.

Final decision:

The manuscript needs major changes.

I hope that my contributions will serve to improve this article and the study you propose.

Thank you very much.

Best regards

Author Response

Dear Editor and reviewers, thank you for your time and effort dedicated to improving our work. We have incorporated all the suggestions you have made and clarified those aspects you have commented on to make it easier for readers to read.

You will find the answers to your suggestions in this document and your additions in blue in the manuscript.

Thank you again.

Reviewer 3

I would like to thank the editors and authors for the opportunity to review the article "Reactive attachment disorder and its relationship to psychopathology: A scoping review"

Treatment approaches aim to repair and strengthen the attachment bond between the child and caregiver to mitigate the impact of early trauma on the individual's psychological well-being. 

The chosen theme is interesting and of greater importance.

In this sense, it is important to map the scientific evidence on the relationship between Reactive Attachment Disorder (RAD) and the presence of psychopathology in children and adolescents and determine the existence of differences in the presence of internalizing and externalizing psychological problems between the RAD group and groups with other disorders or with typical development.

In general terms, I can say that the article presents specific references in interest, however most of them in the introduction and justification of the study are more than 5 years old (more than 77%).

The chosen theme is interesting and of greater importance.

I will now offer my contributions or suggestions for improving the manuscript:

Title/ Abstract

It does not identify a systematic review in the title.

It is not clear whether the authors want to carry out a scoping review (as mentioned in the title), or a systematic review (as mentioned in the abstract). I think it will be an oversight in the title description. Authors must review the title or method referred to in the text.

Response: Thanks for the suggestion, we have changed the title.

  1. Introduction

It is suggested that authors make an explicit statement of all objective(s) or question(s) the review addresses, expressed in terms of a relevant question formulation framework.

This situation must be clear both in the summary and in the Introduction.

Response: the objectives and hypotheses of the study have been modified in an attempt to make them clearer (page 2).

  1. Materials and Methods

2.1. Search strategy

Sugere-se integrar no manuscrito um quadro de População, Intervenção, Comparador, Resultado (PICO) ou uma de suas variantes, para indicar as comparações que serão feitas.

Response: We apologise for the inconvenience but we believe that this aspect is covered at the beginning of section 2.1 (page 3). The study was approached with the PICO strategy in mind. 

It is advisable to review the equations boolean, as some seem to lack parentheses in the Boolean conjugation when "OR" is used.

For example, Where is it described,

"...in Scopus (reactive AND attachment AND disorder) AND (behavior* OR psychological AND health OR psychological AND well-being)

I think the authors wanted the following:

"...in Scopus ((reactive AND attachment AND disorder) AND (behavior* OR psychological) AND (health OR psychological) AND well-being))

I would ask the authors to confirm and to see the following Boolean equations.

Response: We thank the reviewer for this suggestion. The wording of the boleaba operation on page 3 has been changed.

I miss the authors, in point 2.3. Procedure, indicate which the method for synthesizing the studies: for example, whether it was meta-analysis or narrative summary for quantitative data or meta-aggregation/meta-synthesis for qualitative data.

Response: This information has been included in the section 2.3. "Subsequently, the narrative summary was used as a method for synthesising the studies, taking into account that the heterogeneity between studies was too high for meta-analysis."

  1. Results

In point 3.3. Association between RAD diagnosis and other personal, social and mental health difficulties.

The authors describe "The 18 articles included in this review mainly address the presence of internalising, externalising and social difficulties in children with RAD (Table 1)."

It seems to me that the authors wanted to refer to 25 articles and not 18. I ask to confirm.

Response: Thank you, this aspect has been modified. 

  1. Discussion

The discussion reads like a general summary of the results.

I suggest that the authors present the results but contextualize them with the existing literature, demonstrating the importance and contribution of the topic and results to clinical practice and new investigations.

Response: The discussion has been completely revised, taking into account this suggestion (pages 16-19).

The authors discuss the limitations of the studies found. However, they need to discuss the limitations of the systematic review process, for example, they opted for articles only in Spanish and English.

Response: The limitations of the study have been expanded, thanks for the suggestion (page 18).

Conclusion

I miss a chapter with a conclusion in which the authors present the main conclusions in response to the question of research, and value the contributions of the work in relation to the area in question.

I think that at the end of the discussion they form part of the conclusion of the work, I would suggest moving this last paragraph of the discussion to the conclusion.

Response: The conclusions have been expanded. Thank you for the suggestion.

References

Some of the references should be reviewed as some do not comply with the journal's standards.

Most references are more than 5 years old (more than 77%). It is suggested that more recent references be included in the introduction and discussion.

Response: The references in the introduction have been updated where possible. However, we would like to point out that there may be citations from the period 2010 to 2023 as the articles in the review are from that time period. Also, a number of earlier articles have been cited from references such as Bowlby and relating to earlier DSMs.

Thank you for the suggestion, we believe our work is more up to date now.

Final decision:

The manuscript needs major changes.

I hope that my contributions will serve to improve this article and the study you propose.

Thank you very much.

Best regards

Round 2

Reviewer 1 Report

Comments and Suggestions for Authors the authors made appropriate changes to their manuscript.
The authors also clarified some steps, results or statements following the suggestions provided.
the text appears clear and complete

Reviewer 3 Report

Comments and Suggestions for Authors

I would like to thank the editors and the authors for the opportunity to review again the article "Reactive attachment disorder and its relationship to psychopathology: A systematic review"

In general, I can confirm that the authors have re-drafted the manuscript in response to my suggestions for improvement.

I honestly believe that the manuscript in its current situation meets the conditions for publication.

Therefore, my current proposal is to ACCEPT it for publication.

I would like to congratulate the authors for the work done and for their professional behaviour.

Best regards